# Research on the Corn Stover Image Segmentation Method via an Unmanned Aerial Vehicle (UAV) and Improved U-Net Network

**Xiuying Xu, Yingying Gao, Changhao Fu, Jinkai Qiu and Wei Zhang ***

College of Engineering, Heilongjiang Bayi Agricultural University, Daqing 163319, China; xuxiuying@byau.edu.cn (X.X.); 18636869026@163.com (Y.G.); fuchanghao0521@163.com (C.F.); jinkai2020_2023@163.com (J.Q.)

\* Correspondence: zhang66wei@126.com; Tel.: +86-152-4611-0999

**Abstract:** The cover of corn stover has a significant effect on the emergence and growth of soybean seedlings. Detecting corn stover covers is crucial for assessing the extent of no-till farming and determining subsidies for stover return; however, challenges such as complex backgrounds, lighting conditions, and camera angles hinder the detection of corn stover coverage. To address these issues, this study focuses on corn stover and proposes an innovative method with which to extract corn stalks in the field, operating an unmanned aerial vehicle (UAV) platform and a U-Net model. This method combines semantic segmentation principles with image detection techniques to form an encoder–decoder network structure. The model utilizes transfer learning by replacing the encoder with the first five layers of the VGG19 network to extract essential features from stalk images. Additionally, it incorporates a concurrent bilinear attention module (CBAM) convolutional attention mechanism to improve segmentation performance for intricate edges of broken stalks. A U-Net-based semantic segmentation model was constructed specifically for extracting field corn stalks. The study also explores how different data sizes affect stalk segmentation results. Experimental results prove that our algorithm achieves 93.87% accuracy in segmenting and extracting corn stalks from images with complex backgrounds, outperforming U-Net, SegNet, and ResNet models. These findings indicate that our new algorithm effectively segments corn stalks in fields with intricate backgrounds, providing a technical reference for detecting stalk cover in not only corn but also other crops.

**Keywords:** corn stover; UAV; semantic segmentation; U-Net model; attention mechanism

## 1. Introduction

Returning straw to the field is a highly effective measure in conservation tillage due to its ability to prevent environmental pollution and resource wastage caused by straw burning [1]. Moreover, it contributes significantly to maintaining soil moisture, improving soil structure, increasing organic matter content, and promoting robust crop growth. The practice of incorporating straw into the field plays a pivotal role in environmental protection and the sustainable development of modern agriculture. Under suitable conditions regarding soil, water, air, and temperature, adopting conservation tillage methods with straw incorporation can enhance surface activity in the soil while simultaneously reducing weed growth [2,3]. During the process of returning corn straw to the field, numerous nutrients are generated that can partially substitute chemical fertilizers [4]. This not only ensures enhanced crop yields but also benefits soil environment improvement by decreasing the reliance on fertilizers [5–7]. Determining the appropriate coverage rate of straw is an important consideration when determining no-tillage levels and subsidies for straw incorporation [8]. Excessive residue left in fields may have adverse effects on germination rates for soybean seedlings during subsequent seasons. Furthermore, growers commonly perceive returning straw as a time-consuming and labor-intensive task. Therefore, it is difficult to promote straw returning. To actively encourage growers to participate in straw

returning, Heilongjiang Province (in northern China) has implemented a land protection policy [9], which includes a subsidy policy based on the measurement standard of the straw coverage rate. In this context, the detection and supervision of straw incorporation into the field have been put in place. The precise identification of straw morphology in the field can serve as a foundation for the swift assessment of corn straw coverage. This plays a pivotal role in facilitating straw return to the field.

The traditional methods of detecting the straw coverage rate primarily rely on manual techniques, such as the pulling method [10], and the manual estimation calculation method [11]. The traditional testing methods, however, heavily rely on manual execution, leading to low testing efficiency and high expenditure costs. Consequently, this has a detrimental impact on the overall development of the straw coverage testing industry. These methods were time-consuming and labor-intensive. The employment of UAV low-altitude remote sensing (UAV-LARS) technology offers many benefits for crop information collection, including enhanced aerial coverage, accurate and real-time data acquisition, and rapid data transmission [12]. It is widely used to detect crop coverage [13–16]. With the increasing popularity of machine vision recognition technology, the use of image-processing methods to identify straw and detect straw coverage has become a prominent research area. Gausman et al. [17] first used multi-spectral scanner (MSS) images to distinguish between the spectral characteristics of soil and straw. Muhammad et al. [18] utilized multi-temporal satellite remote sensing data from Landsat-7 (ETM+) and Landsat-8 (OLI/TIRS) to assess straw coverage. The R2 value, which represents the correlation between the predicted and actual measured values of straw coverage, was found to be 0.86. Precision agriculture demands the accurate delineation of crop fields at an agricultural scale, a significant challenge for satellite remote sensing due to data quality, atmospheric interference, and modeling methods. Despite its advantages, such as extensive coverage and high efficiency, these constraints limit satellite remote sensing, resulting in low accuracy.

The traditional pull-string method and manual estimation are time-consuming and laborious in detecting straw coverage, while the timeliness of satellite remote sensing images is inadequate and the resolution of the extracted images is insufficient for accurate discrimination between straw and soil. Therefore, it can be concluded that all of these methods have inherent limitations.

To address these problems, researchers used conventional image-processing techniques for straw recognition. Yu et al. [19] aimed to address the issue of straw recognition accuracy being affected by illumination and soil color. They proposed a method to recognize straw-covered images by using the support vector machine (SVM) algorithm and a grayscale transformation function. This method can easily misclassify soil in the background as straw due to the selection of parameters in the prediction function duringSVM training. Li et al. [20] utilized a combination of the fast Fourier transform (FFT and SV algorithms to achieve the automatic recognition of straw coverage, resulting in an average error rate of 4.55%. Wang et al. [21] combined the Sauvola and Otsu algorithms to address the issue of the inaccurate recognition of straw images caused by thin straws. They also conducted field experiments. The experimental results show that the error rate of conventional straw coverage detection is 3.6%. Although traditional image processing methods can address issues such as illumination, soil, and other factors in straw image recognition, they are susceptible to inconsistencies in collection angles and inaccurate image capture. Refurbished techniques could not isolate chopped straw from an intricate rural backdrop.

Although these methods have shown improvement compared to traditional manual detection methods and satellite remote sensing technology, there are still other challenges. For instance, theSV andFFT algorithms generally lack robust generalization ability and struggle to accurately identify finely chopped straws in complex backgrounds. On the other hand, the thresholding algorithm Otsu can effectively segment straw; however, it is susceptible to variations in acquisition angle and equipment, necessitating feature engineering for proper characterization. Manual feature extraction is time-consuming and fails to capture high-level image features, thereby limiting its accuracy.

In recent years, scholars in the field have employed various devices to capture images of straw and conducted research on detecting straw coverage using deep learning techniques. For example, the enhanced U-Net model proposed by Ma et al. [22] addresses the challenges of light interference, soil color changes, and shadow effects in traditional straw identification image recognition. They employed vehicle-mounted monitoring equipment to collect straw images. At the same time, the feature pyramid network is utilized to enhance the extraction of straw features in the image, resulting in an average crossover ratio of 84.78% for the straw dataset; however, the images of the straw collected by the camera on the locomotive will be distorted due to inconsistent shooting angles. Zhou et al. [23] applied the UAV-LARS technique to capture images and proposed to segment straw images using an improved ResNet18-U-Net semantic segmentation model. The algorithm achieves an average segmentation accuracy of 93.70%, but the images are affected by the problem of imprecise cropping. Liu et al. [24] proposed a large-scale collection of straw images using UAV. They utilized the multi-threshold differential gray wolf optimization algorithm to enhance the Precision of straw coverage detection, resulting in a straw coverage detection error rate of less than 8%; however, the algorithm has high requirements for image quality and hardware equipment, making it only suitable for detecting large areas of straw coverage. It can be inferred that the utilization of diverse devices for acquiring straw data and the implementation of deep learning algorithms to detect straw coverage significantly enhance the accuracy of segmentation. Nevertheless, there exist certain drawbacks, such as the influence of inconsistent angles during equipment collection on straw segmentation and its limited applicability to other field sizes.

In summary, based on an analysis of existing research on straw coverage detection and image segmentation methods, it can be concluded that traditional image segmentation algorithms have certain limitations in terms of accuracy. Moreover, they are susceptible to the influence of lighting conditions and low-resolution image acquisition, which hinders the accurate extraction of soil and straw from complex backgrounds. On the contrary, deep learning algorithms demonstrate high accuracy and enable the automatic extraction of intricate features from straw images even under challenging backgrounds. The utilization of receptive field and weight-sharing techniques also reduces the number of parameters during model training. UAV platforms offer new opportunities for the efficient detection of straw coverages at a high throughput rate. By leveraging UAV low-altitude remote sensing technology, independent image acquisition is possible regardless of shooting angles while ensuring speediness. The combination of these technologies enables improved and expedited detection capabilities for assessing straw coverage effectively. Therefore, this study employs UAV low-altitude remote sensing technology along with introducing deep learning to achieve effective detection results for measuring straw coverage accurately. Furthermore, through an extensive search conducted on related studies, no articles focusing on detecting straw coverage in conservation tillage using UAVs or VGG19 with U-Net models were found, thus highlighting the significance and novelty of the topic addressed by this study.

The primary objective of this study is to accurately extract corn stover from the captured images to achieve the precise segmentation of fine straw. Additionally, it aims to assess the quality of straw segmentation and provide a more accurate basis for measuring the degree of no-tillage field return in conservation tillage and determining subsidies for straw field return. Building upon previous research, this paper utilizes UAVs as a data acquisition platform, with a specific focus on corn fields as the research subject. High-definition visible-light images of corn stover covers are captured, followed by data enhancement and other operations to obtain ultra-high-resolution stover maps. Furthermore, a novel image segmentation method for field corn stover is proposed. This method enhances the U-Net model through incorporating transfer learning and replacing the encoder part with an attention mechanism, thereby improving recognition and segmentation ability for fine straw. The Focal-Dice Loss function is selected, along with network optimization techniques, to achieve the cost-effective and highly accurate recognition and segmentation of field corn

stover and fine straw. These advancements offer new perspectives on segmenting and extracting corn stover in large fields while also providing valuable reference material and technical support for improving efficiency in detecting straw coverage during protective farming practices.

## 2. Materials and Methods

### 2.1. Experimental Materials and Image Acquisition

2.1.1. Experimental Materials

The experiment was conducted within the Science and Technology Park of Jianshan Farm, which is located in Heihe City, Heilongjiang Province, China. The coordinates are 48°46′55″ N, 125°19′53″ W. Jianshan Farm is situated in the heart of the internationally renowned Songnen Plain, which is acknowledged as one of the three major black soil belts. The farm covers 400 square kilometers, with 385 thousand acres of arable land primarily used for cultivating corn, soybeans, and dwarf sorghum. The previous crop at the experimental site was corn, which was planted in mid-May 2022 and harvested in early October of the same year, leaving behind a significant amount of straw residue. The imagery was captured using a DJI Phantom 4 multispectral unmanned aerial vehicle equipped with a six-channel multispectral camera. The vehicle was flown at a height of 5 m above ground level. To further investigate the influence of straw coverage on the emergence rates of soybean seedlings, data collection for corn straw was carried out from 23 April to 28 April 2023.

2.1.2. Image Acquisition

The acquired images encompass six bands: red (R), green (G), blue (B), near-infrared (NIR), visible light (RGB), and red-edge (RE). In this research, the visible spectral image was selected as the benchmark data and mounted on the UAV flight platform. The UAV flight parameters were configured using DJI GS Pro (v2.0.17) software. An overlap rate ranging from 65% to 90% was established, which is directly proportional to the Precision of image stitching. Consequently, the forward- and side-phase overlap rates were set at 80%. The ground sampling distance is 0.44 pixels/cm, the shooting speed is three m/s, and the vehicle follows an "S" route. In the isometric shooting mode, the gimbal constantly maintains a vertical position relative to the ground, ensuring the quality of data collection using this method.

To address the compromised image capture caused by lighting obstructions, we have selected a time frame from 10:00 to 13:00, during which abundant sunlight is available. The reason for this is that Heihe City is situated in the northern region of China, characterized by a high latitude, which consequently leads to a relatively delayed sunrise and strong wind intensity during the morning and evening hours. Shifting the testing time earlier would result in inadequate illumination for UAV data collection. Conversely, delaying the testing time may expose unfavorable meteorological conditions, such as ascending thermal air currents and heightened wind strength in the afternoon. These factors could potentially compromise UAV flight stability, thereby impacting image clarity and data quality adversely. Henceforth, we have opted for a specific timeframe encompassing clear weather conditions, ample daylight, and wind speeds below level 3 to ensure optimal data collection. There would be ample sunlight during this time, and the photography angle would be directly perpendicular to the ground surface. The experimental area for data collection is illustrated in Figure 1. By employing UAV for aerial photography, Figure 2 below illustrates a segment of the captured straw images.

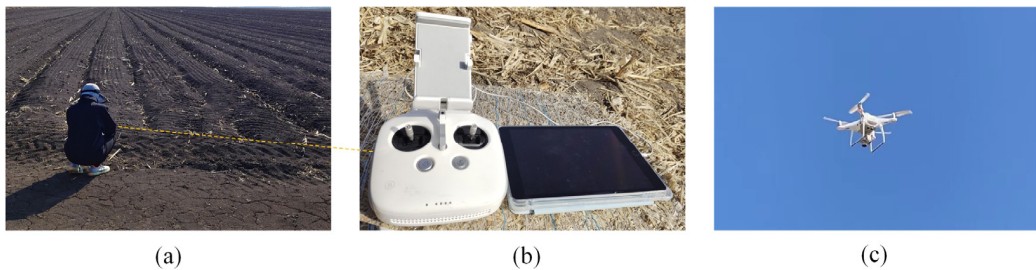

(a)                                  (b)                                  (c)

**Figure 1.** Map of straw data collection in the experimental area. (**a**) Pilot area collection efforts; (**b**) drone remote control with flat screen display; and (**c**) UAV with camera on board shooting vertically on the ground.

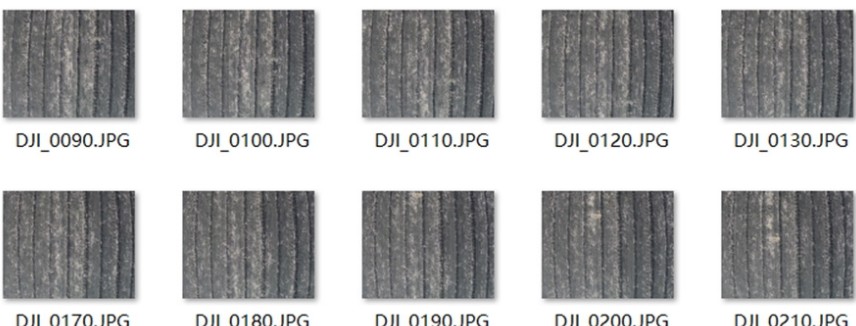

DJI_0090.JPG    DJI_0100.JPG    DJI_0110.JPG    DJI_0120.JPG    DJI_0130.JPG

DJI_0170.JPG    DJI_0180.JPG    DJI_0190.JPG    DJI_0200.JPG    DJI_0210.JPG

**Figure 2.** Visible spectral images of partially collected straw.

### 2.2. Image Preprocessing

The direct utilization of raw image data cannot accurately extract the distinctive features of straw morphology in the field. Therefore, pre-processing operations must be performed on the original images. The image format captured by the UAV platform is JPG, with an original resolution of $1600 \times 1300$ pixels. In cases where the overlap rates of the heading and side phases are excessive and the pixel value is low, direct image recognition may result in the inaccurate extraction of straw information. The UAV is susceptible to wind speed and other factors that can influence its forward movement, resulting in changes in the shooting angle of the image.

Consequently, it is crucial to screen the valid data from the captured images and perform pre-processing operations on this data to ensure the accuracy of straw information extraction. The UAV captured 3500 images, processed using Agisoft Metashape Professional (v2.0.4) software via the specified overlap rate. Camera alignment was performed concerning ground control points, and feature matching was executed on adjacent images. Seamless stitching was ultimately achieved after generating a 3D point cloud, digital elevation model (DEM), and orthophoto data. Figure 3 illustrates the flowchart for stitching visible spectrum images, while Figure 4 displays a map of some of the stitched straw plots.

After the image is spliced, it will be larger than the actual experimental area. It requires manual cropping to remove surrounding silhouettes as well as interference backgrounds, such as utility poles, and ultimately preserve the compelling image within the experimental area. The pixel value of the spliced image is $14{,}228 \times 14{,}147$, which falls under ultra-high resolution. This exceeds the size requirements of the semantic segmentation network for sample data. Consequently, direct resizing may compromise image quality. Furthermore, considering GPU memory consumption, the image must be cropped to the size required by the network for efficient deep learning model training, i.e., $224 \times 224$. The straw image is cropped using the same process and slicing tool in PhotoShop (v20.0.10) software. Figure 5 illustrates the process of cropping the straw image.

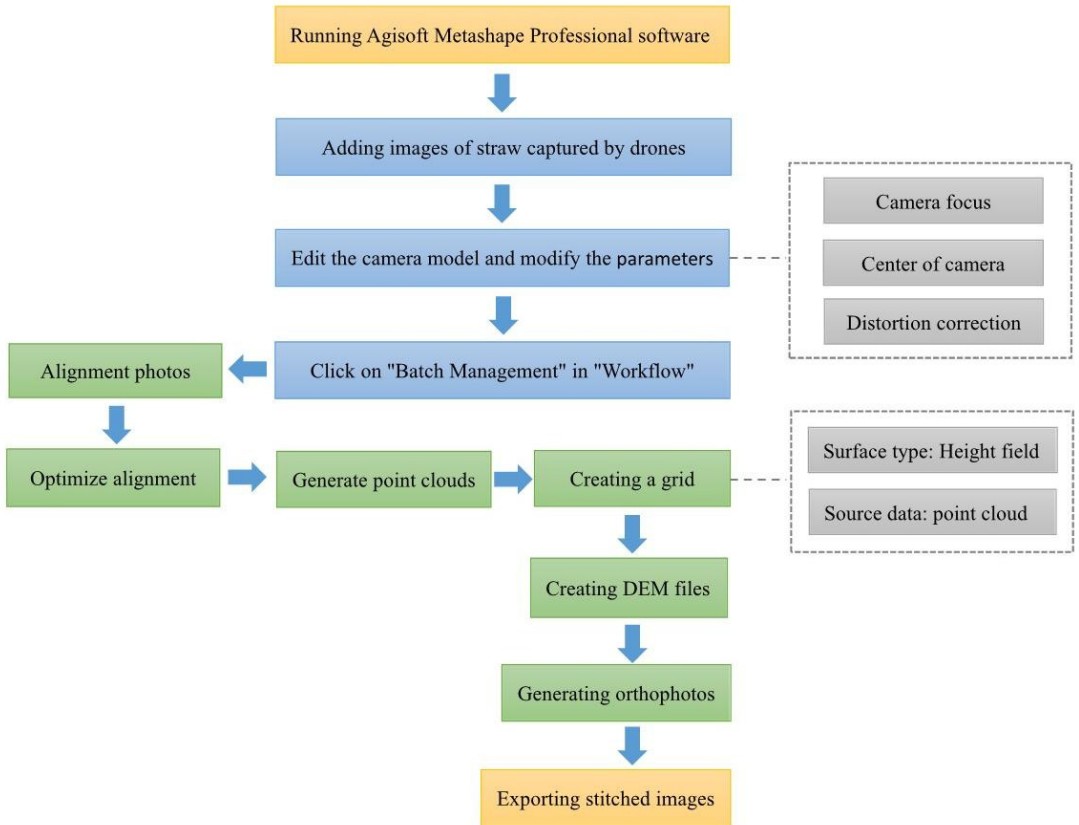

**Figure 3.** Flowchart of visible spectral image stitching.

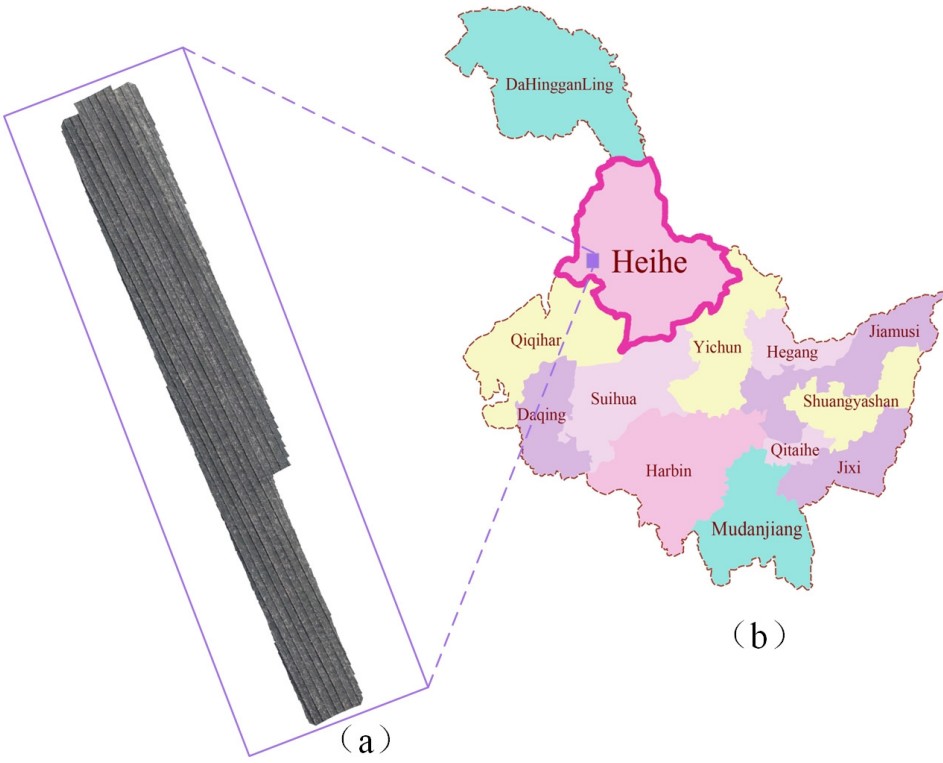

**Figure 4.** Map of some straw plots after stitching. (**a**) Test ground fast after splicing; (**b**) geographic location of the pilot area.

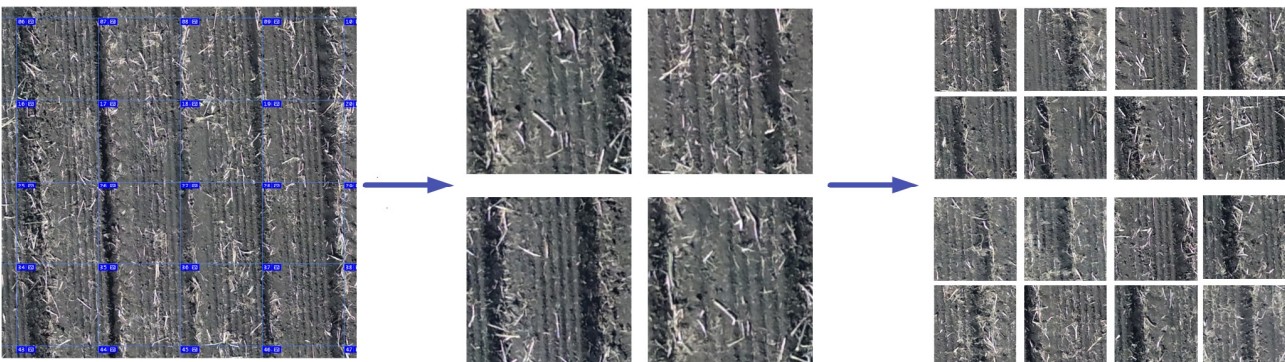

**Figure 5.** Slicing operation.

*2.3. Corn Stover Dataset Production*

Before training the semantic segmentation model, it is crucial to label the target region in the dataset at the pixel level. Consequently, Labelme (v4.5.13) software, compatible with semantic segmentation, is employed to annotate the preprocessed straw data in a "point" manner. The annotated image serves as the ground truth for evaluating the performance of the U-Net semantic segmentation model. The segmentation task of this experiment is categorized into two classes: straw and background. Black denotes the background, while red represents the straw in the image. Upon the completion of labeling, the straw dataset is saved in a JSON file format. To facilitate subsequent training, the dataset is transformed into mask and binary images, both in a JPG format. Figure 6 illustrates the process of creating the dataset.

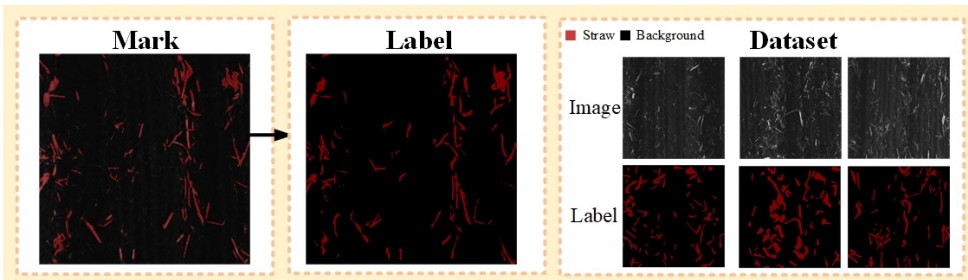

**Figure 6.** Dataset production process.

This paper used the VOC 2007 [25] format to train the network model, and the data size is 256 × 256. Based on the dataset division criteria proposed by Liu et al. [26], the straw data were divided according to the ratio of 6:2:2, i.e., the training set accounts for 60%, the validation set for 20%, and the test set for 20%. Out of 1600 straw data, there are 960 in the training set, 320 in the validation set, and 320 in the test set. Table 1 shows detailed information about the dataset category, image size, division ratio, and number of samples.

**Table 1.** Detailed information on dataset categories, image size dimensions, segmentation ratios, and their sample sizes.

| Dataset Category | Division Ratio | Sample Size |
| --- | --- | --- |
| Training set | 60% | 960 |
| Validation set | 20% | 320 |
| Test set | 20% | 320 |

## 3. Corn Stover Splitting Model

*3.1. U-Net Model*

The U-Net architecture (Figure 7), which exhibits symmetry [27], offers several advantages. Firstly, it effectively addresses the issue of positional information loss in deep-level

pixels for simple samples. Secondly, it enhances boundary confidence more efficiently. During the process of straw coverage detection, interference from factors such as tree shadows, human shadows, and soil can impact the segmentation results of straw images and subsequently affect the accuracy of training subsequent straw segmentation models.

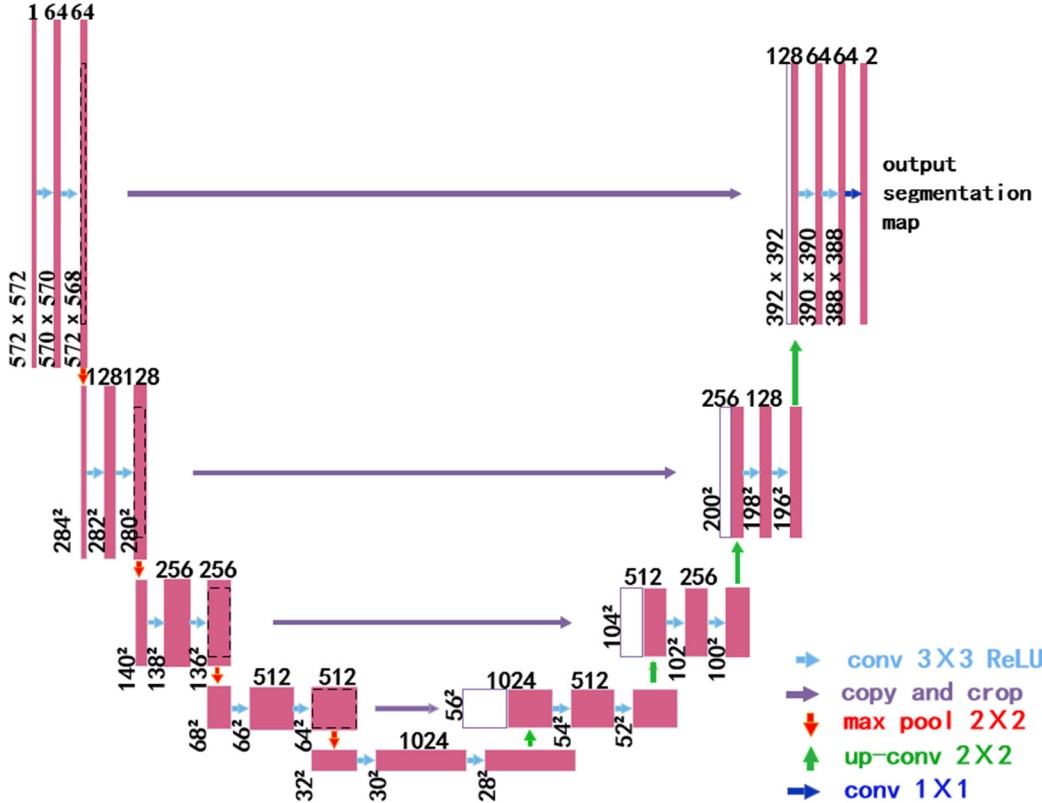

**Figure 7.** U-Net structure diagram.

Long et al. [28] proposed a fully convolutional neural network (FCN). The FCN model has certain limitations, such as the absence of detailed information and spatial correlation among pixels. To address these issues, various solutions have been proposed by researchers, including U-Net [29], SegNet [30], and ResNet [31]. Among them, U-Net stands out as an enhancement over the FCN [32]. Ronneberger et al. [33] proposed the U-Net model in 2015, and its structure is shown in Figure 7 below.

The U-Net is composed of five layers and incorporates four modules: up-sampling, skip connections, down-sampling, and concatenation. In the diagram, the purple arrows indicate that the feature maps in the down-sampling layer are duplicated and cropped before being connected to their corresponding up-sampling layers. The left encoder is responsible for extracting local features from the input data. Each down-sampling module consists of two convolutional operations with activation [34], followed by max pooling to effectively construct them. This process is iterated four times to proficiently extract and classify data features. The skip connections are utilized to integrate the up-sampled feature maps with the down-sampled ones through precise cropping and channel-wise concatenation during the up-sampling process. Among these connections, the decoder on the right side of the network represents the expansive path where extracted features are effectively fused. This facilitates the prompt restoration of the original size and yields segmentation results with an enhanced resolution. Given that each down-sampling operation leads to a loss of edge features, it becomes imperative for the decoder component to concatenate features extracted from the encoder to minimize this detrimental effect.

### 3.2. Improved U-Net Model

To effectively segment the delicate cornstalks in images with complex backgrounds, four aspects are optimized, namely the encoder, attention mechanism, loss function, and network optimization.

U-Net is similar to SegNet, ResNet, and VGG in terms of the encoder part, all of which extract image features from the convolutional layer; however, U-Net's simplicity makes it suitable only for small datasets and insufficiently sensitive to the intricate edges of straw in complex field backgrounds, thus being unable to effectively handle large datasets of finely chopped straw samples. The VGG19 network, trained on the large-scale dataset ImageNet, can now extract image features in complex backgrounds. This network consists of 19 layers, typically 16 convolutional layers and 3 fully connected layers. The fully connected layers have 4096, 4096, and 1000 neurons, respectively, with a softmax classifier performing prediction probability. Fully connected layers have more neurons connected than those in the convolutional layers [35], leading to the increased consumption of computational resources when training the network, along with an increased difficulty and duration of model training, resulting in phenomena such as insufficient memory and overfitting.

To tackle the challenges associated with the U-Net coding module in intricate environments, the low accuracy of straw segmentation, and the inability to effectively segment the edges of the straw, we incorporated migration learning [36] by replacing the first five layers of the VGG19 [37] network with the U-Net coding module. This model-based migration is accomplished by fusing the source and target domains and adjusting the model parameters. In our study, we accelerated the feature extraction process of straw by loading the weights of the pre-trained VGG19 parameters as the initial parameters for the straw segmentation model training, thereby enhancing the model's generalization performance [38–41].

The aforementioned enhanced U-Net network necessitates the further extraction of fine straw location information. Subsequently, a CBAM [42] convolutional attention mechanism module is incorporated behind each convolutional layer within the coding segment. Four operations are executed to re-weight significant straw features within the image, augment the feature fusion of the U-Net network, and boost the secondary extraction capability of straw image features, thereby enhancing the segmentation performance of finely chopped straw edge details. CBAM comprises two attention mechanisms, namely the channel attention module (CAM) and spatial attention module (SAM), which are sequentially connected. Compared to a separate CAM and SAM, it superiorly mines the edge feature information of fine stover in corn stover under a complex backdrop. Figure 8 illustrates the CBAM convolutional attention mechanism module.

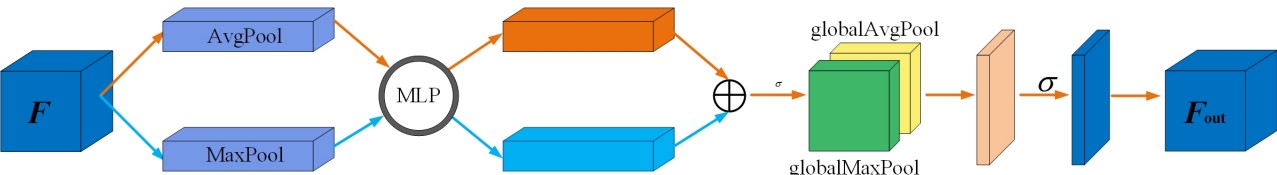

**Figure 8.** CBAM: convolutional attention mechanism module.

After the global maximum pooling (MaxPool) and global average pooling (AvgPool) of the input feature map, F, within the CAM, the resultant feature maps are, respectively, fed into a multilayer perceptron (MLP). Subsequently, the weights of each channel output from the MLP are multiplied with the input feature map, channel-by-channel, to generate a feature map, F1, containing channel information weights. This processed feature map, F1, is then integrated into the SAM. The CAM-weighted feature map is ultimately employed to generate two compressed feature maps, F2, by employing both MaxPool and AvgPool, superimposing F1 and F2, and finally outputting the feature map, Fout, after a convolution operation, which incorporates both channel and spatial information weights.

During the training process of the semantic segmentation model, the loss function is usually used to evaluate and monitor the model [43,44]. Straw segmentation is essentially a

pixel-level binary classification task, and this paper proposes the Focal-Dice Loss function due to the fact that the ordinary loss function can only focus on segmentation performance or discrimination performance alone, and cannot converge effectively [45]. This being the case, the loss function can focus on the model's ability to discriminate between straws and the segmentation ability of straws, which in turn solves the problem of sample classification imbalance. The Focal Loss is used as an auxiliary loss function by dynamically changing each target's Dice Loss weight and cross-entropy loss weight to deal with difficult-to-segment and challenging-to-classify fine straw effectively. The Focal Loss function can be described as follows:

$$Focal\ Loss = \begin{cases} -(1-p)^{\gamma}log(p), y = 1 \\ -p^{\gamma}log(1-p), y = 0 \end{cases},$$ (1)

where $\gamma$ denotes the attention parameter, $p$ is the predicted value of the positive sample straw, and $y \in \{0, 1\}$ is the accurate straw label.

When the pixel value of the foreground region is lower than the pixel value of the background region during training, the network will prioritize the background region, which causes the problem of losing some satisfactory straw information. Therefore, Dice Loss is the primary loss function with which to improve the segmentation accuracy of small-scale delicate straws. The Dice Loss function can be expressed as follows:

$$Dice\ Loss = 1 - \frac{2 \times \sum_{I=1}^{N} pi \times gi}{\sum_{i=1}^{N} pi^2 + \sum_{i=1}^{N} gi^2},$$ (2)

where $N$ is the total number of pixel points, $pi$ is the value of the ith pixel predicted by the model, and $gi$ is the value of the ith pixel of the actual label.

The gradient of the improved U-Net network trained based on the above operations was too smooth, and the overall network needed to be optimized to continue improving the model's straw segmentation ability. Set the size of all convolution kernels in the model to $3 \times 3$ with a step of one. Adjust the input image size to a three-channel image of $224 \times 224$ and perform a five-stage convolution-pooling operation. With the same size, multiple small convolution kernels are used to be stacked to improve the depth and effectiveness of the network to some extent [46]. Figure 9 demonstrates the improved VGG19-UNet encoder network structure.

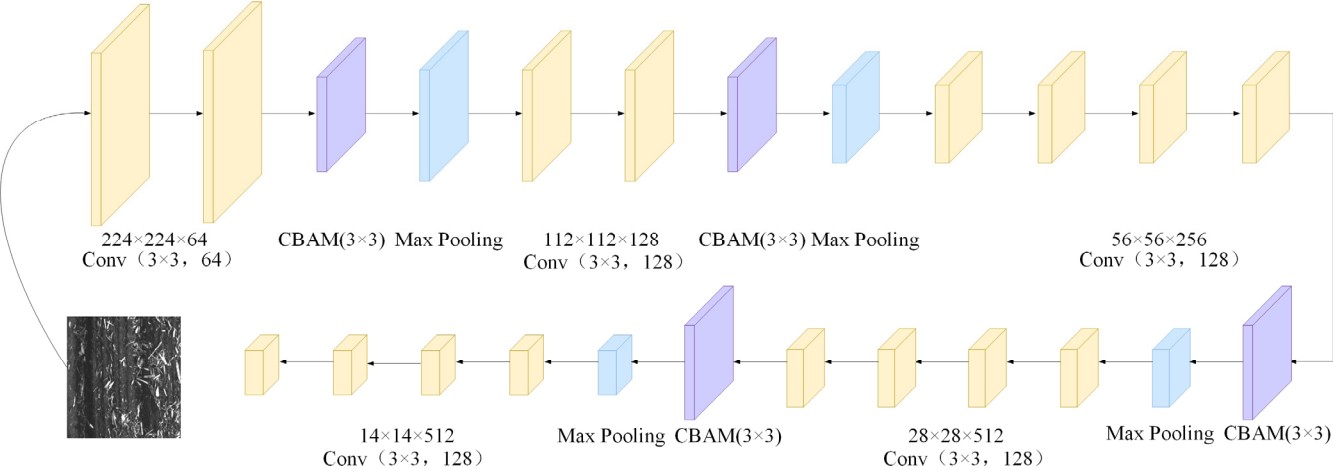

**Figure 9.** Improved VGG19-UNet encoder network structure.

## 4. Experiments and Results

*4.1. Model Training*

4.1.1. Development Environment

The experimental development environment is Windows 11 (64-bit), using Pytorch 1.7.1 as the deep learning framework; the main dependency libraries are Random, Opencv, Numpy, PIL, etc.; and the Python version and CUDA are 3.7 and 11.0, respectively. The parameters of the hardware equipment are an Intel Core i5-1200H CPU (16 GB of RAM), an NVIDIA GeForce GTX 3050 GPU, and 4G memory.

4.1.2. Parameter Settings

The sample data size affects the segmentation accuracy of the model. In order to verify this effect, three sets of straw images with different sizes are set up, these being $128 \times 128$, $256 \times 256$, and $512 \times 512$. The number of iterations (it/s) and Pixel Accuracy (PA) will be used as the evaluation indices. Table 2 represents the comparison results of the straw segmentation model with different sample data sizes.

**Table 2.** Comparison results of different sample data sizes on straw partitioning models.

| Model | Sample Size (px) | Iterations (it/s) | PA (%) |
|---|---|---|---|
| | $128 \times 128$ | 18.57 | 82.07 |
| U-Net | $256 \times 256$ | 12.32 | 85.34 |
| | $512 \times 512$ | 3.42 | 83.99 |
| | $128 \times 128$ | 20.39 | 84.69 |
| SegNet | $256 \times 256$ | 12.09 | 88.92 |
| | $512 \times 512$ | 4.10 | 87.01 |
| | $128 \times 128$ | 24.95 | 88.43 |
| Our algorithm | $256 \times 256$ | 13.35 | 93.87 |
| | $512 \times 512$ | Max out memory | / |
| | $128 \times 128$ | 23.54 | 85.48 |
| ResNet | $256 \times 256$ | 6.33 | 90.62 |
| | $512 \times 512$ | 4.95 | 88.73 |

It can be seen that when the sample data input size is $512 \times 512$, the processing speed of the four straw segmentation models is slow, an our algorithm has the problem of bursting the video memory. When the sample data input size is $128 \times 128$ due to the original low pixel value of the image, the accuracy obtained is also lower than that of the segmentation results when the size is $256 \times 256$; therefore, the segmentation results obtained using a sample data size of $256 \times 256$ are ideal. The number of iterations per time for processing data using U-Net, ResNet, and SegNet models was 12.32 it/s, 6.33 it/s, and 12.09 it/s, respectively. Our algorithm's speed has increased to 13.35 iterations per second.

The same straw data ($256 \times 256$) were input each time to ensure that the model had the same reference in each run. During training, the model was trained using the Adam optimizer with an initial learning rate, Init_lr, of 0.0001 and a batch size of input samples per iteration of 2. The total training period epoch was set to 100 rounds to prevent the model from generating overlearning. Focal Loss and Dice Loss were used as loss functions to straw partition the dataset. The network training process is divided into two phases: First, in the freezing phase, the weight parameters of the encoder part are frozen from 0 to 49, and only the decoder part is trained. The maximum training period of the freezing phase is ten rounds [47], and a more significant learning rate is used at the beginning to allow the model to quickly jump out of the local minimum of the loss function. Next, in the unfreezing phase, the encoder part is unfrozen from 50 to 99, and the whole model is trained. The model is fine-tuned during training using a smaller learning rate until the model converges.

4.1.3. Evaluation Indicators Comparison Model Setup

In order to better verify the effectiveness of the model in recognizing straw, mIoU, PA, Precision, and Recall are used as the evaluation indices of the algorithm in recognizing the target. The above four indices take the value in the interval of [0, 1], and the higher the value, the better the segmentation result represented:

$$mIoU = \frac{1}{k+1} \sum_{i=0}^{k} \frac{P_{ii}}{\sum_{j=0}^{k} P_{ij} + \sum_{j=0}^{k} P_{ij} - P_{ii}}, \tag{3}$$

$$PA = \frac{TP + TN}{TP + TN + FT + FN}, \tag{4}$$

$$Precision = \frac{TP}{TP + FP} = \frac{TP}{N}, \tag{5}$$

$$Recall = \frac{TP}{TP + FN}, \tag{6}$$

In this study, $k$ denotes the number of categories, $pij$ predicts $i$ to be $j$, $pji$ predicts $j$ to be $i$, and $pii$ predicts $i$ to be $i$;

$TP$ denotes the number of voxels correctly categorized as straw; $FP$ denotes the number of voxels incorrectly categorized as straw; $TN$ denotes the number of voxels correctly categorized as the background; and $FN$ denotes the number of voxels incorrectly categorized as the background.

*4.2. Ablation Experiment*

The U-Net model was employed as the baseline network to facilitate a comprehensive comparison of the impact of each enhanced module. Through the controlled variable approach, incremental improvements were incorporated into the model's architecture. Subsequently, training on the designated dataset enabled an evaluation of their effectiveness based on accuracy metrics in the validation set. Table 3 presents the encoder section's performance for straw segmentation across different network structures. Notably, VGG19 outperformed other feature extraction networks when integrated with U-Net, achieving an mIoU of 81.93% and a PA value of 92.66%. This superiority can be attributed to VGG19's larger number of trainable parameters compared to the U-Net, ResNet18, and ResNet50 networks. Consequently, it excels at extracting intricate straw features from complex backgrounds and enhancing the overall PA value for straw segmentation.

**Table 3.** Accuracy of different network structures for straw segmentation in the coding phase.

| Feature Extraction Network | mIoU/% | PA/% |
|---|---|---|
| U-Net | 73.54 | 85.34 |
| ResNet18 + U-Net | 79.07 | 89.88 |
| ResNet50 + U-Net | 79.41 | 90.03 |
| VGG19 + U-Net | 81.93 | 92.66 |

The U-Net model was enhanced by incorporating various modules, and, subsequently, five groups of ablation tests were conducted to specifically evaluate the impact of each module on improving straw segmentation. Table 4 presents a comparative analysis of the results obtained after integrating different modules.

The first experiment represents the original U-Net network. In the second experiment, the encoder part of U-Net is replaced with VGG19. Experiment 3 incorporates the CBAM module based on experiment 2. Experiment 4 selects focal dice based on experiment 3. After each experiment, the model's mIoU for the straw segmentation and PA of each network structure on the validation set is recorded.

**Table 4.** Comparison of straw splitting results after adding different modules.

| Number | VGG19 | Encoder CBAM | Focal-Dice Loss | mIoU/% | PA/% |
|--------|-------|--------------|-----------------|--------|------|
| 1 |  |  |  | 73.54 | 85.34 |
| 2 | ✓ |  |  | 81.93 | 92.66 |
| 3 | ✓ | ✓ |  | 82.75 | 93.52 |
| 4 | ✓ | ✓ | ✓ | 83.23 | 93.87 |

The original U-Net network achieved an mIoU of 73.54% for straw segmentation, as indicated in Table 4; however, this study demonstrates a significant enhancement in the mIoU for straw segmentation by incorporating the VGG19 network and CBAM into the architecture, resulting in an impressive increase of 9.69 percentage points to reach 83.23%. This improvement can be attributed to the enhanced capability of the network in extracting straw features. Comparative experiments 2 and 3 demonstrate that the CBAM effectively integrates semantic information from deep feature maps with spatial information from shallow layers, thereby mitigating the impact of interference (such as soil) on straw segmentation results and further enhancing the network's feature extraction capability. Consequently, there is an improvement of 0.82 percentage points in the mIoU compared to when the CBAM is not utilized. Comparative experiments 3 and 4 demonstrate that incorporating Focal Loss with the dice function significantly enhances the mIoU of the network model by an increase of 0.48 percentage points compared to excluding focal dice. By selecting Focal-Dice Loss as the improved model's loss function, it not only emphasizes global features in straw images but also effectively addresses the issue of imbalanced sample classification caused by similar colors between straw and soil, paying attention to finer local features. Consequently, it more accurately extracts boundary information from feature maps.

### 4.3. Comparison Model Setup

This paper compares several benchmark models of U-Net, SegNet, our algorithm, and ResNet. The U-Net and SegNet methods are the classical image segmentation methods. Both Our algorithm and ResNet use the migration learning method to replace the encoder part of U-Net with improved VGG19 and ResNet50, respectively, and all of these models are encoder–decoder structures. Table 5 represents the specific compositions of the encoders and decoders in these four models.

**Table 5.** Comparison of different model structures.

| Model | Encoder | Decoder |
|-------|---------|---------|
| U-Net | Ten convolutional blocks | Deconvolution linked to an encoder |
| SegNet | VGG16 | Inverse convolution for multi-layer feature fusion |
| Our algorithm | VGG19 | Same as U-Net |
| ResNet | ResNet50 | Same as U-Net |

### 4.4. Comparison and Analysis of Results

The feasibility of the model is analyzed based on the curves of the mIoU and loss, as depicted in Figure 10, which illustrates the variations in the mIoU and loss values at different iterations for the trained model. A smaller loss function indicates the superior performance of the model. From Figure 10a, it can be observed that, during the training process, the U-Net model exhibits significant instability, with multiple steep slopes in its mIoU values. In comparison, ResNet demonstrates relatively stable mIoU values compared to SegNet; however, it does not converge to a stable state until after 140 iterations of training. On the contrary, our algorithm achieves both higher and more consistent mIoU values compared to the other three models. Particularly noteworthy is that, from 0 to 80 iterations of training, our algorithm displays an upward trend followed by a gradual

leveling off between 80 and 200 iterations, indicating its accurate segmentation capabilities surpassing those of other models. As evident from Figure 10b, U-Net takes approximately 100 iterations to initiate convergence and smoothen out its performance. Similarly, ResNet also encounters convergence issues within its initial hundred rounds of training. Although SegNet briefly outperforms our algorithm in terms of loss during rounds 60–90, overall our algorithm performs admirably well and demonstrates good convergence among all four models throughout training sessions. After around 125 rounds of training, our algorithm's loss value gradually converges and remains steady over subsequent iterations—indicating close prediction proximity to actual values with minimal deviation and showcasing its robust adaptive performance.

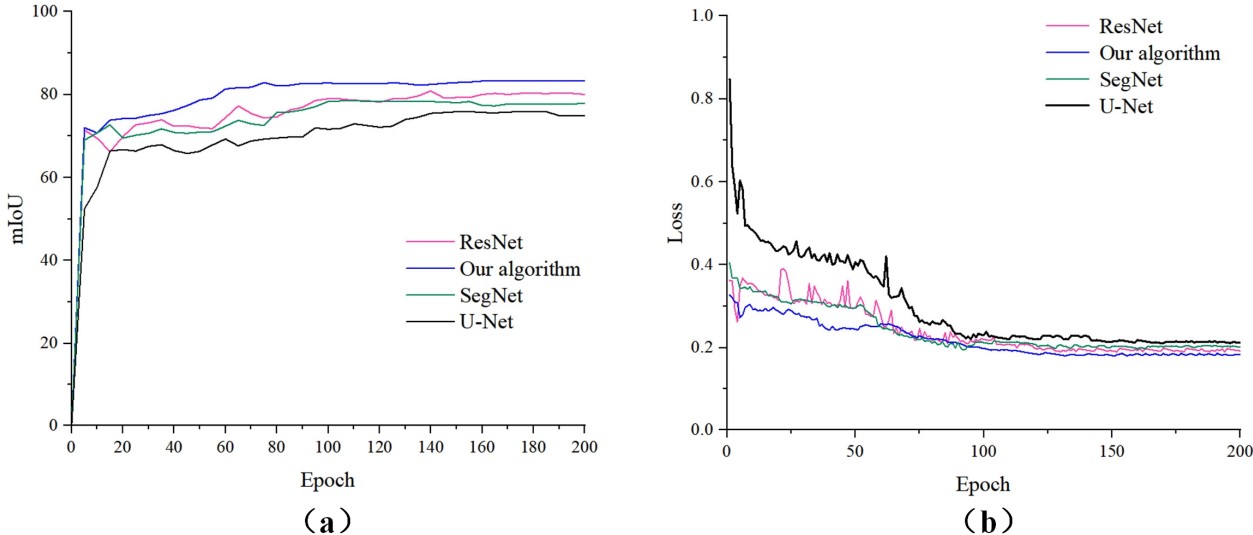

**Figure 10.** Loss and mIoU variation curves for four models during training. (**a**) mIoU; (**b**) Loss.

Table 6 represents the segmentation results of U-Net, ResNet, our algorithm, and SegNet, concerning straw and background.

**Table 6.** Comparison of segmentation results of four models regarding straw and background.

| Model | mIoU/% | Recall/% | Precision/% | PA/% |
|---|---|---|---|---|
| U-Net | 73.54 | 82.96 | 83.11 | 85.34 |
| SegMet | 77.86 | 86.06 | 88.47 | 88.92 |
| Our algorithm | 83.23 | 90.66 | 91.98 | 93.87 |
| ResNet | 80.02 | 89.75 | 90.11 | 90.62 |

To further evaluate the performance of network enhancements in segmentation, we replaced the encoders of three distinct classification networks, namely ResNet, our algorithm, and SegNet, with that of the U-Net. The segmentation outcomes of these networks were then compared to those of the original U-Net. Table 6 presents the segmentation results of various models for both straw and background.

The original U-Net segmentation yields the least satisfactory results, with an average intersection and merger ratio of 73.54%. This can be attributed to the high complexity involved in extracting straw features from the farmland background in our dataset. The simple U-Net model fails to adequately capture detailed straw features in images. Additionally, Figure 11 demonstrates challenges in segmenting fine straws, as there are instances of breakage observed. In deep learning, a combination of linear and nonlinear calculations is required to extract individual features effectively. As the network becomes deeper, its ability to extract straw features improves significantly due to ResNet's utilization of a depth-first search strategy and residual connectivity. The PA achieved a remarkable 93.87%, while the mIoU reached an impressive 83.23%. Moreover, our algorithm demonstrated exceptional

Precision of 91.98% and Recall of 90.66% during training. This significant enhancement in performance can be attributed to the strategic deepening of network layers through encoder replacement, which effectively enhances the segmentation capability of the network. By incorporating advanced techniques such as the CBAM and Focal-Dice Loss, our model focuses on both the global and temporal extraction of intricate straw features, thereby mitigating gradient disappearance caused by increased network depth and accelerating convergence speed during model training. Additionally, leveraging small convolution kernels in network optimization facilitates the precise capture of fine features, enabling the more accurate extraction of local details about field straw information for improved accuracy in straw identification and segmentation tasks. Our novel algorithm surpasses the traditional U-Net, SegNet, and ResNet models in terms of its superior performance in straw segmentation and recognition.

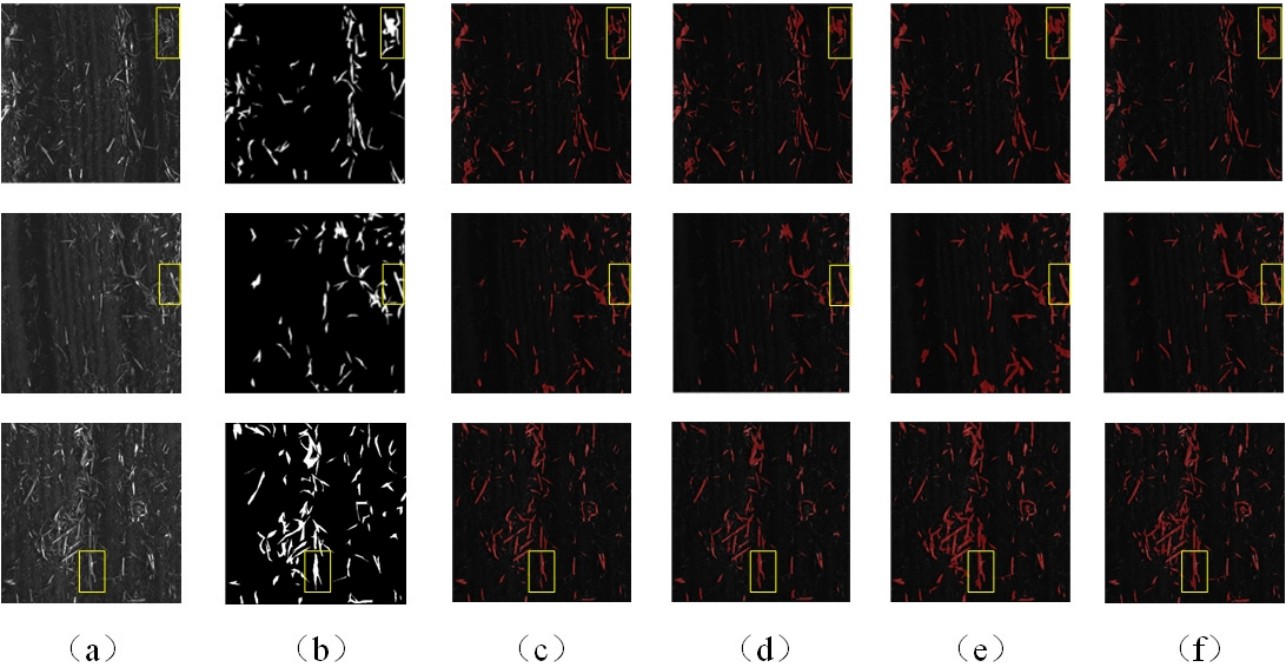

**Figure 11.** Partial test results of four models for straw splitting. (**a**) Original image; (**b**) label image; (**c**) U-Net; (**d**) SegNet; (**e**) ResNet; and (**f**) our algorithm.

In the test dataset, 100 images of cornstalks were randomly selected. These images were then segmented using the four different models. Figure 11 represents some of the test results of the four models for stover segmentation, from which the effect of stover segmentation can be more intuitively seen. The yellow box represents a partially boxed image that focuses on the fine straw segmentation in a single image. It aims to provide more intuitive results from different models' segmentations. Comparing Figure 11c–f with Figure 11b, it can be observed that, in the first row of yellow boxes, Figure 11c only achieves the simple identification of the straw contour compared to Figure 11b. Moreover, some fine straw's edge contour information is not adequately extracted, resulting in the ineffective accurate segmentation of the straw image. On the other hand, Figure 11d demonstrates improved segmentation results compared to Figure 11c. For instance, within the center of the yellow boxes, there is a ratio greater than 1.0% between the middle part and the background in Figure 11d, while this ratio exceeds 2% at the same location within the yellow boxes. Additionally, closer examination reveals even higher ratios between certain parts and backgrounds within these yellow boxes when compared to the original labeled images.

However, there is still an issue of under-segmentation when identifying fine straws on the left side of the image. The straw segmentation results are presented in the Figure 11, where Figure 11e,f resemble those in Figure 11b, but with noticeable differences. Specifically, Figure 11e suffers from overfitting and fails to eliminate interference from other back-

grounds such as soil during straw segmentation, resulting in obvious over-segmentation. In contrast, Figure 11f can more accurately segment fine straws from background noise; within the yellow box, its segmentation results closely match both labeled and true values. Moving on to the second and third rows, we observe similar phenomena to those discussed earlier. These findings demonstrate that our proposed algorithm for straw segmentation outperforms three other models compared in this paper when used for corn stover images. It achieves the accurate extraction and segmentation of stalk boundaries even under complex background conditions due to several improvements made to our algorithm for this task. By comparing different models' results for straw segmentation, we conclude that our algorithm shows feasibility regarding improvement.

Moreover, various industries and fields encounter challenges with data segmentation accuracy, such as cracks and small cells. The enhancement techniques discussed in this article for the U-Net algorithm can be similarly applied to areas including crack detection, medical image segmentation, and fruit object segmentation, among others.

### 5. Conclusions

The identification and segmentation of corn stalks in the field are crucial for meeting the technical requirements of reduced tillage and qualifying for subsidies related to returning crop residues. Numerous researchers have utilized machine learning, deep learning, and other techniques to extract features from images; however, there is a lack of research focusing on using deep learning techniques to segment fragmented stalks. Most existing methods for stalk segmentation suffer from certain drawbacks that hinder the accurate identification and segmentation of fragmented stalks. The latest research conducted by scholars [22] involved the utilization of onboard cameras and deep learning algorithms for straw coverage detection. To enhance the capability of extracting straw features, they replaced the U-Net encoder part with a ResNet34 structure. On a dataset consisting of straw images captured by onboard cameras, our algorithm achieved an mIoU performance of 84.78%, which is comparable to the results obtained in their study on straw segmentation; however, during the process of dividing fine straw, the camera acquisition angle is easily influenced by the forward movement of the locomotive, resulting in a substantial number of non-farmland scenes. Consequently, this hampers the effectiveness of extracting straw data and necessitates the subsequent manual removal of irrelevant background elements, such as the sky and ground, leading to a significant workload. In a study conducted by scholars [23], images capturing farmland straw under conservation tillage were obtained using a low-altitude UAV. They developed an improved U-Net (ResNet18-UNet) semantic segmentation algorithm for straw cover without incorporating any additional optimization modules. Nevertheless, this algorithm proved ineffective in accurately distinguishing fine straw from soil amidst complex backgrounds and failed to achieve accurate extraction with an mIoU performance segmentation rate of 81.04%. Our algorithm surpasses theirs. Therefore, based on a comprehensive review of scholarly research, we propose a novel U-Net straw segmentation method. Our innovative approach to straw segmentation encompasses several key aspects: Firstly, unlike conventional data acquisition devices such as satellite remote sensing, airborne cameras, or handheld cameras that are susceptible to factors like shooting angles, this study employs low-altitude remote sensing technology UAVs for the precise and high-resolution collection of straw data in farmland. Secondly, previous studies have utilized methods such as BP neural networks and threshold segmentation for segmenting collected straw data within the realm of deep learning models; however, these approaches suffer from lengthy training times and limited generalization capabilities when it comes to accurate straw segmentation. In contrast with these methodologies, our paper leverages the U-Net model as the foundation network while integrating the latest VGG19 network into the encoding section of the U-Net model to reduce training parameters and incorporate CBAM modules. This not only enables our algorithm to focus on global features but also significantly enhances its ability to extract fine-grained straws from complex backgrounds. Thirdly, prior segmentation models merely specified input

data sizes without comparing their impact on model training results across different sizes. To establish universally applicable data sizes for various models, this study selects three distinct dimensions: $128 \times 128$ pixels, $256 \times 256$ pixels, and $512 \times 512$ pixels. Furthermore, the influence of these dimensions on segmentation accuracy during the model training process is thoroughly examined and discussed. Fourthly, all convolutional kernel sizes in our model are set at a size of $3 \times 3$, which optimizes network design further, enhancing both the generalization ability of our model and the accuracy of straw segmentation.

The original U-Net model is enhanced in four aspects: the encoder, attention mechanism, loss function, and network optimization. This algorithm effectively segments and extracts fragmented stalks from complex field images. The main conclusions are as follows:

(1) To address the issue of the inaccurate segmentation of straw images caused by shadows and the challenges in identifying fragmented straws due to varying shooting angles during data collection, we propose utilizing a transfer learning approach. This method replaces the encoding phase of the original U-Net architecture with the first five layers of the VGG19 backbone network. Additionally, we integrate the CBAM convolutional attention mechanism and optimize the entire network using the Focal-Dice Loss function. This enhancement focuses on segmenting fine-grained straw edge details, reducing parameters and computational complexity, and improving the accuracy of segmenting corn straws in complex backgrounds.

(2) During the training process, the model demonstrates the expected performance when the input data size is $256 \times 256$. Compared to the other three algorithms, this algorithm demonstrates superior performance in corn stalk segmentation tasks. Consequently, our algorithm is suitable for segmenting the stalks of various crops and other tasks where there is minimal distinction between the foreground and background.

(3) The straw segmentation algorithm advanced in this study fulfills the processing requirements for capturing aerial images of straw coverage; however, it is exclusively designed for the low-altitude, small-scale, and high-precision detection of straw coverage, making it unsuitable for large-scale detection.

(4) Our algorithm serves as a technical reference for detecting the straw coverage rate of corn and other crops in the field, providing valuable insights to enhance the U-Net model. It constitutes a valuable resource for assessing the straw coverage rate of crops while also providing innovative ideas to enhance the U-Net model.

The challenges encountered in the research on straw coverage detection using UAVs can be summarized as follows: Firstly, ensuring measurement accuracy is crucial for accurately obtaining the distribution and density of straw, which directly impacts the accuracy of detection results; however, the current precision of UAVs in monitoring straw coverage remains limited. Different UAV devices exhibit varying levels of precision, thus necessitating improvements in UAV detection accuracy and the reduction in measurement errors. Secondly, dataset annotation plays a vital role in ensuring data quality by precisely labeling straw. Before model training, effective data processing and analysis are necessary for the straw data collected from UAVs. Currently, manual annotation continues to dominate dataset construction but consumes significant time and resources during data processing. Extracting valuable information from large volumes of raw data, establishing sound data analysis models, as well as achieving fast transmission and real-time analysis are urgent issues that need to be addressed. Thirdly, handling complex scenarios poses challenges in detecting and predicting straw coverage rates due to factors such as occlusion, lighting variations, and complex backgrounds encountered in actual agricultural environments; therefore, the further refinement and optimization of relevant algorithms are required.

The article proposes the application of UAVs and deep learning technology for the recognition and segmentation of corn stalk images in fields. Although our proposed algorithm has been experimentally demonstrated to segment stalk images effectively, there are still certain limitations. Future research and enhancements can be pursued in the following directions: We will continue to enhance the model. Despite achieving a test set accuracy of 93.87% and reducing the average coverage error to 0.35%, there is still

room for further improvement in terms of precision. Our algorithm necessitates high-performance hardware devices for execution, and without GPU acceleration the running efficiency requires enhancement. To achieve more comprehensive and accurate straw coverage detection, we will integrate multimodal data by combining multispectral or thermal infrared images obtained from UAVs with traditional visible-light images.

**Author Contributions:** Conceptualization, X.X. and Y.G.; methodology, X.X., Y.G. and J.Q.; validation, X.X., Y.G. and C.F.; formal analysis, Y.G. and X.X.; investigation, X.X.; resources, X.X.; data curation, X.X.; writing—original draft preparation, Y.G.; writing—review and editing, X.X.; visualization, X.X. and Y.G.; supervision, X.X.; project administration, X.X.; funding acquisition, W.Z. All authors have read and agreed to the published version of the manuscript.

**Funding:** This research was funded by the China Agriculture Research System of MOF and MARA, (grant number: CARS-04-PS32); the Technical Innovation Team of Cultivated Land Protection in North China (grant number: TDJH201808); Platform Construction of Protected Tillage Technology Research Center in Heilongjiang Province (grant number: PTJH202102); and the Key Laboratory of Soybean Mechanized Production, Ministry of Agriculture and Rural Affairs, China.

**Institutional Review Board Statement:** Not applicable.

**Data Availability Statement:** The data presented in this study are available on request from the corresponding authors.

**Acknowledgments:** The authors would like to thank the editors and anonymous reviewers for their constructive comments and suggestions.

**Conflicts of Interest:** The authors declare no conflicts of interest.

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
