# Peer review of "Research on the Corn Stover Image Segmentation Method via an Unmanned Aerial Vehicle (UAV) and Improved U-Net Network"

_agriculture, doi:10.3390/agriculture14020217_

Round 1

Reviewer 1 Report

Comments and Suggestions for Authors

-- The authors lack a summary of existing literature to facilitate the presentation of the innovative aspects of this paper. Such information should be included to facilitate readers to know your paper’s innovation compared with previous similar research regarding the method design.

--From Figure 10 (b), it appears that the training did not exhibit stable convergence. This raises concerns about the reliability of the test results.

--The ablation study has not been made in the manuscript.

-- The experiments are insufficient. More advanced compared methods and experimental results are needed.

--The analysis is too basic, and it would be more beneficial to incorporate data for more in-depth analysis and discussion.

----In the conclusion section, the authors should try to compare your findings with the respective outputs from recently published research efforts. What have other studies that examined and assessed different algorithms found and discuss whether their outcome is in agreement or not with your main conclusions.

Author Response

Thank you very much for your valuable time and effort in reviewing the revised version of the manuscript. The comments of the review report are very valuable for increasing the quality of our manuscript, and also provide significant guidance for our future research. We have carefully revised the manuscript according to reviewer’s comments, hoping to get your positive reply.

Attached to this letter is our point-by-point response to the comments made by the reviewer.

Thank you again for allowing us to resubmit the revised manuscript.
Best regards,
The authors

Reviewer 2 Report

Comments and Suggestions for Authors

The presented paper is about “Corn Stover Image Segmentation Method by 2 UAV and Improved U-Net Network”. The overall evaluation is sufficient and the general contents can be improved with the following suggestions:

1.       Please explain more about the innovation of this research

2.       Please mention the cleared practical objectives of the research

3.       Please compare the results with the finalized research and industrial systems.

4.       One very important note is the “comparison of research with existing challenges in the industry”. Could you solve some? Can you cite?

5.       The experimentation was done in Heihe City, Heilongjiang Province, China. Can you expand and include the same system in other cities in China? How about other countries? How? Why?

6.       The experimentations were done in a time frame from 10:00 to 13:00. Don’t you think, it can be a disadvantage of the developed system when you want to apply it in industry / real farms in daily use?

7.       Some general modifications are needed: some figures have insufficient resolution (Ex: fig 7), and some minor grammatical revisions are required.

8.       Presenting some suggestions for future studies is appreciated.

Comments on the Quality of English Language

Minor revision required

Author Response

(The authors gave the same response as above.)
